# IMproving the practice of intrapartum electronic fetal heart rate MOnitoring with cardiotocography for safer childbirth (the IMMO programme): protocol for a qualitative study

Guillaume Lamé,[1] Elisa Liberati,[1] Jenni Burt,[1] Tim Draycott,[2,3] Cathy Winter,[2,3] James Ward,[4] Mary Dixon-Woods[1]

¹The Healthcare Improvement Studies Institute (THIS Institute), University of Cambridge, Cambridge, UK
²School of Social and Community Medicine, University of Bristol, Bristol, UK
³Women and Children's Health, North Bristol NHS Trust, Westbury on Trym, UK
⁴Engineering Design Centre, University of Cambridge, Cambridge, UK

**Correspondence to**
Guillaume Lamé;
guillaume.lame@thisinstitute.cam.ac.uk

## ABSTRACT

**Introduction** Suboptimal electronic fetal heart rate monitoring (EFM) in labour using cardiotocography (CTG) has been identified as one of the most common causes of avoidable harm in maternity care. Training staff is a frequently proposed solution to reduce harm. However, current approaches to training are heterogeneous in content and format, making it difficult to assess effectiveness. Technological solutions, such as digital decision support, have not yet demonstrated improved outcomes. Effective improvement strategies require in-depth understanding of the technical and social mechanisms underpinning the EFM process. The aim of this study is to advance current knowledge of the types of errors, hazards and failure modes in the process of classifying, interpreting and responding to CTG traces. This study is part of a broader research programme aimed at developing and testing an intervention to improve intrapartum EFM.

**Methods and analysis** The study is organised into two workstreams. First, we will conduct observations and interviews in three UK maternity units to gain an in-depth understanding of how intrapartum EFM is performed in routine clinical practice. Data analysis will combine the insights of an ethnographic approach (focused on the social norms and interactions, values and meanings that appear to be linked with the process of EFM) with a systems thinking approach (focused on modelling processes, actors and their interactions). Second, we will use risk analysis techniques to develop a framework of the errors, hazards and failure modes that affect intrapartum EFM.

**Ethics and dissemination** This study has been approved by the West Midlands—South Birmingham Research Ethics Committee, reference number: 18/WM/0292. Dissemination will take the form of academic articles in peer-reviewed journals and conferences, along with tailored communication with various stakeholders in maternity care.

## INTRODUCTION

Preventable harm related to childbirth can be catastrophic for women, children and families,[1] as well as causing high costs for health systems.[2] One important source of preventable harm in maternity care arises

### Strengths and limitations of this study

► A multidisciplinary team of obstetricians, social scientists, midwives and engineers will collaborate to characterise the technical and social mechanisms that may affect the safety of electronic fetal heart rate monitoring (EFM) in labour.

► The study combines the strengths of ethnographic research and engineering approaches to systems analysis and risk assessment.

► This project will generate a detailed characterisation of the errors, hazards and failure modes in intrapartum EFM and will help to inform the development of an intervention that will directly target the reasons for problems in interpretation and response to cardiotocography traces.

► Three maternity units across the UK will be selected; the generalisability of the findings will require careful assessment.

from sub-optimal fetal heart rate monitoring, particularly electronic fetal heart rate monitoring (EFM) using cardiotocography (CTG) in labour.[3] Effective interventions to improve the practice of EFM have remained elusive, perhaps in part because of a lack of sound understanding of its range of influences on safety. We aim to generate a comprehensive characterisation of the technical and social mechanisms that may affect the safety of EFM in labour with the goal of informing the development of a targeted intervention for improvement.

### Fetal monitoring in labour

Two principal methods can be used to monitor the fetal heart rate in labour: intermittent auscultation and EFM with CTG. National Institute for Health and Care Excellence (NICE) guidelines recommend offering intermittent auscultation to women

at low risk of complication during labour[4]; EFM is the recommended option in the presence of certain signs or conditions specified in the guidelines (such as fresh vaginal bleeding, hypertension or high temperature or when oxytocin is used).[4] Our study focuses on the use of CTG, where the baby's heart rate is monitored through a Doppler ultrasound transducer and the woman's contractions are monitored through a uterine pressure transducer. Both signals are monitored continuously and recorded and/or printed as a CTG trace.[5] These traces are then used to detect fetal heart rate abnormalities and trigger appropriate action.

## Interpretation

Interpretation and response to intrapartum CTG traces involve a series of complex sociotechnical processes with many potential points of failure. Interpretation of CTG traces requires healthcare professionals to consider the classification of the trace in the context of the clinical circumstances of the mother, the fetus and the status of labour, in order to formulate a response and take action. The initial classification involves review of four features on the CTG trace: the baseline heart rate, baseline variability, the presence of accelerations and the presence or absence of decelerations, as well as characteristics of variable decelerations if present. NICE guidelines provide criteria to classify each feature as 'reassuring', 'non-reassuring' or 'abnormal'.[4] The trace itself is then classified in one of four ways: (1) 'normal' (all features are reassuring), (2) 'suspicious' (one non-reassuring feature and all other features are reassuring), (3) 'pathological' (one abnormal feature or two or more non-reassuring features) or (4) 'need for urgent intervention' (acute bradycardia or a single prolonged deceleration for 3 min or more).

In determining *responses* to non-normal traces, NICE guidelines provide management indications to be considered in context with the clinical circumstances. The guidelines also recommend documenting any maternal or fetal risk factors, the woman's and the unborn baby's condition, CTG review every hour using a structured document, a 'fresh eyes review of the CTG, and seeking senior advice (from a senior midwife or an obstetrician) when the CTG is difficult to interpret or is not categorised as normal.

Despite the guidance, studies consistently show high levels of interobserver and intraobserver variability in the interpretation of CTGs.[6–9] Some technological solutions have been proposed, including the introduction of computerised decision support systems for CTG interpretation in labour. However, their effectiveness remains unclear: a large randomised controlled trial did not indicate a benefit of computerised decision support.[10] Research on response to non-normal CTG traces has remained underdeveloped.

Overall, training for healthcare staff is currently the most frequently proposed solution to suboptimal CTG practice.[3 11 12] The NHS England Saving Babies' Lives 'care bundle' states that all staff undertaking fetal monitoring should be trained in both the review system and the escalation protocol,[12] with mandatory yearly training and competency assessment. Implementation of this bundle is measured as the percentage of staff who have received training in fetal monitoring; the percentage deemed competent in fetal monitoring and the percentage whom have successfully completed mandatory annual updates.

It is clearly important that such training be supported by high quality evidence. A 2011 systematic review concluded that training for CTG interpretation in labour can lead to improvements in individuals' interpretation skills, interobserver agreement and management of intrapartum CTGs.[13] However, the training interventions included in the systematic review were highly heterogeneous in format and content (including e-learning, case reviews, monthly audit with feedback, voluntary review sessions and clinical supervision through tele-didactics), making it difficult to draw definitive conclusions on what features and mechanisms of the training were linked with practice improvement. The authors of the systematic review also noted that the generally poor quality of the reviewed studies warrants caution with the findings.[13]

Perhaps because so little evidence exists, training programmes are not standardised[14] and where programmes are implemented there are difficulties in demonstrating positive impacts. For example, in Denmark, all midwives and physicians in maternity units were required to take part in a CTG education programme consisting of e-learning, a 1-day course and a final written assessment.[15] The evaluation of this programme suggested that it did not decrease the risk of birth hypoxia.[16] A national intervention in Sweden yielded similar results.[17 18]

One challenge in moving the field forward is that most of the effort so far is based on the assumption that improvement requires targeting deficits in individuals' knowledge.[19 20] Improving each staff member's knowledge and skill is clearly important, but insufficient attention has been granted to the other dimensions of why it may be difficult to improve interpretation and response to EFM. CTG interpretation can, for example, be hindered by cognitive biases: individuals sometimes rely on intuition rather than objective guidelines to interpret and document intrapartum CTGs.[21]

Even when intrapartum CTGs are interpreted correctly, the response may be suboptimal. Social, organisational and cultural features of maternity units and the wider institutions in which they sit may inhibit staff from taking the appropriate action, communicating their concerns or reacting appropriately to a request for intervention.[22 23] Disagreements and divergences between midwives and obstetricians and conflicts over professional boundaries are an unfortunate characteristic of some maternity units.[21 24 25] Multiple other features of the labour process, including pressures on facilities and aspirations of parents for their birth experience make the circumstances of decision-making and mobilisation of response particularly challenging.

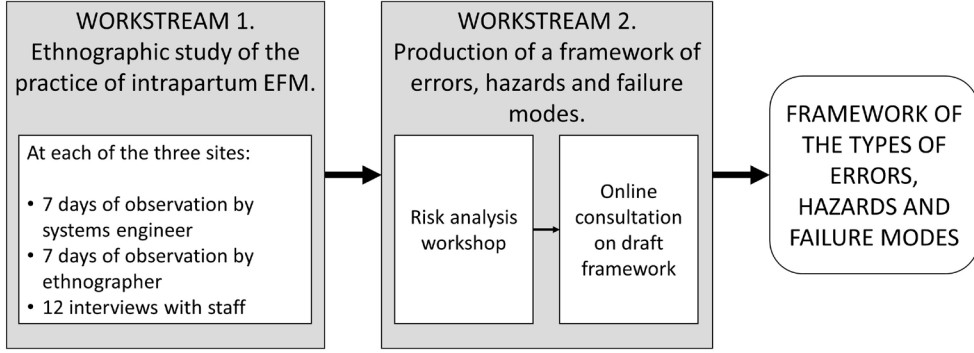

**Figure 1** The two workstreams and the associated research activities in the IMproving the practice of intrapartum electronic fetal heart rate MOnitoring with cardiotocography for safer childbirth (IMMO) study.

Interpreting and reacting to a CTG trace is therefore best understood as a complex sociotechnical process involving individuals from multiple professions and disciplines, taking place over a number of stages and in highly pressurised contexts. Given this, purely technical interventions (eg, computer-assisted CTG analysis) and individual-based training are unlikely to fully address these challenges. We propose that understanding what can go wrong when EFM is used requires full characterisation of the work and social practices involved, the multiple professions who conduct such practices and the context where the process takes place.[26]

## METHODS
### Aims
The overarching aim of this study is to advance understanding of how intrapartum EFM is currently performed in UK maternity units and where risks may occur, in order to inform the development of an intervention to improve practice. The study comprises two workstreams (figure 1):
1. An ethnographic study informed by systems engineering to characterise how intrapartum EFM is currently undertaken in UK maternity units.
2. Production of a framework of the types of errors, hazards and failure modes in intrapartum EFM in UK maternity units.

The study is expected to run between April 2019 and June 2020.

### Workstream 1: an ethnographic study of intrapartum EFM
This workstream adopts an ethnographic approach combined with systems engineering analytical techniques to characterise how intrapartum EFM is currently undertaken in UK maternity units. We will conduct *observations* and *semi-structured interviews* with healthcare professionals in three different maternity units in the UK, in order to:
▶ Map the activities (and relevant risks and hazards) involved in EFM.
▶ Identify and describe the contextual, cultural and sociotechnical factors that influence practices of EFM with CTG.

*Ethnography* is an approach to social and organisational research that draws on researchers' close observation of and involvement with people in a particular setting, with the aim of accessing their *point of view*—namely, their perspectives in and on the world they inhabit.[27] This approach allows the examination of important aspects of clinical work that may be invisible or difficult to articulate by professionals themselves and that may not be amenable to measurement in the traditional sense.[28] It is especially well suited to identifying the informal interactions that may create or prevent risk and to shedding light on the multiple influencing factors that shape clinical practice. Ethnography may also offer insights on the wider organisational and cultural dynamics that may explain why accidents or 'close calls' are welcomed as a learning opportunity in some contexts and ignored or normalised in others.[29]

*Systems engineering* focuses on how to design and manage complex systems over their lifecycles. Adopting 'systems thinking' principles[30] and approaches from human factors analysis,[31] it seeks to ensure that all relevant aspects (social and technical) of a complex process or system are considered and integrated into a whole. The approach is particularly useful in gaining a comprehensive and thorough understanding of the risks and hazards that may affect complex healthcare processes. Systems engineers working on health services can collect and use multiple types of data, both quantitative and qualitative, including data collected through observation.[32–34]

### Eligibility criteria
The research participants in this study will be staff in the maternity units who are directly or indirectly involved in the process of EFM with CTG. We will observe staff in the participating maternity units who are directly or indirectly involved in the process of EFM with CTG, including, for example, obstetricians, midwives, nurses, anaesthetists, maternity care assistants, maternity theatre staff, auxiliary/administrative staff and management staff. Women and birth partners/relatives will not be the main

participants of the study, but are likely to be included in the observations.

For the interviews, we will include doctors and midwives in the participating maternity units who are directly involved in using EFM with CTG. We do not plan to interview women and birth partners/relatives.

## Sampling

Three maternity units will be included in the study. They will be selected purposively based on their size, calculated through annual number of births. Units' size and geographic situation have also been shown to be correlated with CTG interpretation skills.[35] We will recruit one small unit (fewer than 2000 births/year), one medium unit (2000–5000 births/year) and one large unit (more than 5000 births/year). We will also take into account the geographic situation of the hospital. The objective of this sampling strategy is to understand CTG processes and practices and their variations in practice.

In each maternity unit, we will recruit up to 12 individual members of staff for interviews. Interview participants will be selected purposively: we will seek to interview participants with different professional backgrounds (midwives and obstetricians), seniority and professional experiences in maternity care.

## Observations

We expect that a social scientist and a systems engineer will each spend up to 7 days in the three participating maternity units (together or at different times), combining day and night observations and conducting observation blocks of around 8 hours per day/night. The focus of the observations will be the process through which CTG traces are classified and actions are documented; observers may also 'shadow'[36] midwives and obstetricians in order to understand the interactions between professionals and between professionals and parents, and the escalation mechanisms used in response to CTG traces. Our observations will focus on EFM with CTG in intrapartum care (rather than ante-natal care); we therefore expect the bulk of the observations to be conducted in labour wards. However, depending on how each unit organises admission procedures and early labour checks, and because of the practicalities of shadowing staff, observers may also occasionally visit the antenatal ward.

The ethnographers' observations will focus mainly on the social and contextual factors that influence fetal monitoring practice and outcomes. The systems engineer will capture and map the constituting activities and the hazards and risks that characterise the process using human factors concepts drawn from existing frameworks, for example, methods and models to guide observations (eg, the Systems Engineering Initiative for Patient Safety (SEIPS) framework,[37] the Yorkshire framework of factors contributing to incidents in hospitals,[38] process modelling[39] and task analysis[40]).

The data collected will consist of anonymised field notes taken during the observations. At the end of each day, these notes will be dictated and recorded using an encrypted voice recorder for later transcription or written up manually.

No photos or videos will be taken of human subjects, but we may take pictures of equipment. Researchers will additionally request relevant documents (eg, local guidelines, CTG proformas and documentation tools, training materials, posters, etc) from sites. If these documents contain identifying information about any individuals, they will be anonymised prior to storage and analysis.

### Consent for observations

As we have found in previous studies, it will not be practical or appropriate to obtain written consent in all situations where we will be conducting observations. We are conscious that in some circumstances asking people for written consent for observations can make them uncomfortable, disrupt clinical work or alter people's behaviour. In such situations, obtaining written consent is more likely to be for the researchers' benefit than those being studied. Therefore, we plan to use an approach we have used successfully in previous studies, which relies on obtaining permission from those being observed, ensuring that those who wish to opt out can easily make this known and recording only completely anonymised data.

We will ensure that staff being observed are informed of the project and are given written information to explain it. Researchers will always explain who they are and will wear an appropriate identifying badge. They will obtain verbal permission from staff where possible (sometimes this may be from a senior person on behalf of a group) to conduct observations. They will only enter the bed space of pregnant and postpartum women with the permission and agreement of clinical staff and women and will leave immediately if requested to do so, or if there is any indication (even unvoiced) that women or birth partners/relatives would prefer them not to be there.

Women and their birth partners/relatives are not the focus of the study, and we will not seek their written consent. However, the nature of the ethnography means that we may carry out observations of staff as they come into contact with women while carrying out their routine clinical duties. Pregnant and postpartum women will only be observed with their permission and agreement and the permission of clinical staff. Women and birth partners will be advised verbally and in writing (using posters and leaflets) that they can decline observations. The researchers will be sensitive to explaining the aims of the study in a way that will not raise undue concerns in women and birth partners.

### Interviews

The interview schedule will cover participants' experience of EFM, their views on EFM, the training they have received, and their understanding of the factors that may influence EFM processes and outcomes. Staff will be offered the choice of being interviewed individually

or in a small group of two or three participants. Interviews may also be arranged by telephone if participants are not available on the days of the visits. All interviews will be audio-recorded on an encrypted voice recorder (with participants' consent) and transcribed verbatim for analysis.

### Consent for interviews

We will obtain written informed consent for all recorded interviews.

### Data analysis

Data analysis will run alongside ongoing fieldwork and will be conducted in two phases, comprising initial, disciplinary-specific analyses of data, followed by an integrative analysis. In the first phase of analysis, the ethnographers and the systems engineer will analyse their observation data separately (ie, the systems engineer will only analyse data they will have collected, and the ethnographers will only analyse data collected by ethnographers). This is because observations are expected to be dependent on the perspective and sensitising concepts used by the different observers. In this phase, ethnographers and systems engineers will additionally analyse the whole interview dataset separately.

In their respective first phase analyses of observation and interview data, the researchers will adopt different but complementary approaches. The ethnographers' analysis will be based on the constant comparative method.[41] It will be informed by sensitising concepts identified through an earlier literature review, including the role of power and psychological safety. These concepts may be revised, modified or made redundant as analysis proceeds. The engineer's analysis will be based on systems thinking principles and risk analysis approaches. The main aim of the analysis will be to produce a comprehensive description of the process of EFM with CTG, using systems modelling techniques to describe the processes, actors and their interactions.[39] Depending on the nature of the findings, frameworks such as the Yorkshire and SEIPS frameworks may be used to support the analysis.[31 38]

In the second phase of analysis, the researchers will integrate their analyses. This combination of different disciplines and bodies of knowledge, will facilitate a synthesis between a rich understanding of individual sites and the ability to generalise from the specifics of these cases, to enable the development of new knowledge and inform action in this area.

Debriefing sessions of the research team will be recorded, transcribed and treated as data alongside the field notes. QSR NVivo software will be used to aid the coding, management and retrieval of data.

### Workstream 2: building a framework of errors, risks and failure modes in intrapartum EFM

In Workstream 2, we aim to build a framework of errors, hazards and failure modes in EFM with CTG, and identify the underlying mechanisms that can explain these.

In doing this, we will draw on the findings from Workstream 1 as well as concepts from the sociology of risk (eg, normalisation of deviance[42]), psychology (eg, cognitive biases,[43] automaticity,[44] groupthink[45]) and human factors (eg, sociotechnical systems[46]).

### Design, data collection and data analysis

Our approach is informed by recent attempts to integrate different sources of knowledge into risk assessment efforts.[47] In our case, the choice of modelling approaches (eg, process maps, stakeholders maps, risk analysis methods[39 47]) will to some extent depend on the information collected and cannot yet be determined. We plan to combine different approaches, as previous research has shown the benefit of combining complementary risk management methods in health services.[48] First, we will create a representation of the intrapartum CTG interpretation and management process, drawing on observations from the ethnographic study (Workstream 1) and the extant literature. Second, using prospective risk analysis approaches (such as Failure Modes and Effects Analysis[49] or Hierarchical Task Analysis[50]) we will analyse this process to identify where problems may occur. Finally, where relevant, we will link these issues to documented patterns (eg, cognitive biases or normalisation of deviance) in order to build on the existing knowledge of these phenomena and how to tackle them.

As part of our strategy for ensuring multidisciplinary synthesis, we will conduct a 2-day workshop with the research team and relevant experts, as well as other stakeholders in healthcare risk management, obstetrics and midwifery. At this workshop, participants will reflect on, adapt and develop the representation and analysis of the intrapartum CTG interpretation and management process, to inform framework development.

The output of this phase of work will be a theoretically and empirically grounded framework of the errors, hazards and failure modes in the interpretation of, and reaction to, intrapartum CTG traces.

### Consent for workshop participants

Prior to the workshop, participants will be asked to give consent to the recording of the workshop and the use of data produced during the workshop (including anonymised quotes) for research purposes. This is so that we can report on the process of building the framework in publications. To this end, the workshop will be audiorecorded, and the recording will be transcribed.

### Assessing the framework's comprehensiveness through a stakeholder consultation

To ensure that the framework is comprehensive in its description of errors, hazards and failure modes in the process of EFM, we will submit the final product developed from the workshops to the assessment of a broad range of stakeholders, using an online consultation. Participants will be separate from those who participated in the workshop, and will represent obstetrics, midwifery,

risk management, third sector organisations and other relevant groups.

It is too early at this stage to decide on the exact form of this consultation, which will be designed to complement the content and nature of the framework. However, it is likely to comprise a questionnaire on the comprehensiveness of the framework, asking participants to suggest additional items or remove existing ones, and asking them to rate the clarity of each item. It is possible that this may be done through a citizen science approach, using The Healthcare Improvement Studies Institute's platform.

The product of this consultation will be a revised framework, ready for use in the next phases of this research programme, which ultimately aims to develop and evaluate an intervention to improve the use of intrapartum EFM.

### Patient and public involvement (PPI)

Issues with maternity safety in general, and EFM in particular, have received wide attention in recent years. Our exchanges with PPI groups have shown that the issue is of critical importance to pregnant women. While pregnant women are not the primary focus of this study, we are keen to engage and involve this group along with other stakeholders in the design, conduct and dissemination of the research.

We have engaged with a network of women (maternity users) to advise us on how best to introduce the study to women in labour during our ethnographic study. These individuals reviewed our participant material (participant information leaflet, consent form and poster) and modifications were made to the material and to guidance on how and when to approach women in labour.

Our objective at this stage is to understand how professionals make decisions to act on a certain type of clinical information (CTG traces). The opportunities to involve pregnant and postpartum women in the research itself are limited. We plan to engage deeply with women (maternity users) in later stages of this work programme, when we will consider potential interventions to improve the practice of EFM.

### ETHICS AND DISSEMINATION

This study has received ethical approval from the West Midlands—South Birmingham Research Ethics Committee, reference number: 18/WM/0292.

The main risks in this project are likely to arise when researchers are in direct contact with women/relatives in the clinical setting, during the ethnographic study (Workstream 1). We are aware of the sensitive nature of conducting observational research in a maternity care setting and are experienced in the conduct of such work. We will seek to reduce risks in the following ways:

► Making sure that staff, women, and partners/relatives are informed about the project, using information sheets and posters.

► The ethnographic field researchers are highly experienced researchers with extensive expertise in sensitive research. They will provide ongoing support and supervision for the systems engineer while on site. The researchers will always explain who they are and will wear an appropriate identifying badge. They will obtain verbal permission where possible (sometimes this may be from a senior person on behalf of a group) to conduct observations, and staff and women will have the right to refuse to be observed if they wish.

► The researchers will shadow members of staff and will only enter clinical areas such as labour rooms or theatres if this is essential. They will only enter the environment of women and partners/relatives with the permission and agreement of clinical staff and women.

► It is acknowledged that labour can be distressing for women/relatives, particularly if problems arise. The ethnographic field researchers will check with staff and women/relatives if they are happy for them to be present, and will leave immediately if there is any indication (even unvoiced) that staff, women or their families would prefer them not to be there.

► The researchers will take full hygiene precautions.

The findings of this study will be communicated in peer-reviewed journal articles and conferences. We will also consider additional communication channels to convey the results to professionals, for example, blog posts and communication on social media.

**Contributors** GL, EL, JB, TD, CW and MDW conceived the initial idea for the study. GL coordinated the writing of the protocol with input from all co-authors. EL, JB, TD, CW, JW and MDW commented on the protocol and provided substantial ideas in their respective areas of expertise. GL drafted the paper and all authors revised it. All authors approved the submitted version of the paper.

**Funding** This research project is carried out by The Healthcare Improvement Studies Institute at the University of Cambridge. The Institute is supported by the Health Foundation, an independent charity committed to bringing about better health and health care for people in the UK. This work was also supported by MDW's Wellcome Trust Investigator award WT09789. MDW is a National Institute for Health Research (NIHR) Senior Investigator.

**Competing interests** TD is a trustee of the PRactical Obstetric Multi-Professional Training (PROMPT) Maternity Foundation. The PROMPT Maternity Foundation (PMF) is an independent charity (registered charity number 1140557) set up to save the lives of mothers and babies through evidence-based, multi-professional training and research. This includes training on fetal heart rate monitoring. TD and CW are members of the steering group of PMF. CW is seconded from North Bristol Trust as the lead research midwife for PMF.

**Patient consent for publication** Not required.

**Provenance and peer review** Not commissioned; externally peer reviewed.

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
