## [Reviewer comments · BMJ Open]

ARTICLE DETAILS

TITLE (PROVISIONAL)	IMproving the practice of intrapartum electronic fetal heart rate MOnitoring with cardiotocography for safer childbirth (the IMMO programme): Protocol for a qualitative study
AUTHORS	Lamé, Guillaume; Liberati, Elisa; Burt, Jenni; Draycott, Tim; Winter, Cathy; Ward, James; Dixon-Woods, Mary

VERSION 1 - REVIEW

REVIEWER	Ramon Escuriet Catalan Health Service. Government Faculty of Health Sciences. Blanquerna-University Ramon Llull
REVIEW RETURNED	31-Mar-2019

GENERAL COMMENTS	This protocol presents a study as part of a wider research project. the protocol is of high interest for those interested in this topic and willing to improve clinical practice on the field. The manuscript is well written and includes the main relevant aspects for its understanding Methods section. line 59. Authors say "in three maternity units in the UK". This could be confusing, since it can be interpreted as representing all UK maternities (111 according to NHS Maternity Statistics, England 2016-17). It is suggested to change the term "representative" to "different" and to specify that the activity performed in these maternities may reflect the activity set of public maternities with a similar volume of annual deliveries. It is also suggested to authors to include the dates planned for the study. Start date and expected end date.
---

REVIEWER	Valerie Smith School of Nursing and Midwifery, Trinity College Dublin, Ireland
REVIEW RETURNED	09-Apr-2019

GENERAL COMMENTS	Review bmjopen-2019-030271 This is a well-written, clearly described protocol for a novel, interesting study in the area of fetal monitoring in pregnancy. I have some minor comments only which I hope are helpful; 1. Page 3, lines 46-47, under 'Fetal Monitoring in Labour', it might be of benefit to mention, in addition to the two main types of monitoring, when each is used/recommendations for use of each.2. I'm not very clear how the Systems Engineering data are extracted from the ethnographical study, and integrated; suggest insert a few 'plain language' lines to help better explain this (page 7, Lines 22-23 where you state "it seeks to ensure that all relevant
--

	aspects (social and technical) of a complex process or system are considered and integrated into a whole”) 3. Please provide a rationale for why doctors and midwives with < 1 year experience are excluded; especially as you state on page 8; “we will seek to interview participants with different professional backgrounds (midwives and obstetricians), seniority and professional experience.....” Junior level staff could provide valuable data to this study? 4. The study I understand takes place in 3 maternity units; but can you clarify will observations occur in all locations within these units where EFM is likely to take place, or focus on a single location such as labour ward? If it includes all locations within the unit, will your sample for interviews be drawn from across locations within the settings? Please clarify in your protocol 5. In varied places throughout the manuscript the full term ‘electronic fetal monitoring’ is used, whereas, in other places the abbreviated EFM is used (e.g. page 9, lines 17-18); please review for consistency of whichever you chose to use throughout 6. On page 6 you state; “The objective of this sampling strategy is to understand CTG processes and how they vary between units and regions.” Where/how does this fit into your analysis and framework development? 7. References are required for the ‘risk assessment models’ in the following sentence; “The hazards and risks that characterize the process using human factors concepts drawn from existing risk assessment models to guide observations” 8. Where you state; the ethnographers and the systems engineers will analyse the whole dataset separately. Can you clarify is this referring to data that are collected “together and at different times”; i.e. independently gathered data as well as collectively gathered data?
--	--

VERSION 1 – AUTHOR RESPONSE

Reviewer #1 – Ramon Escuriet

Reviewer’s comments	Authors’ response
This protocol presents a study as part of a wider research project. The protocol is of high interest for those interested in this topic and willing to improve clinical practice on the field. The manuscript is well written and includes the main relevant aspects for its understanding	Thank you for your feedback.
Methods section. line 59. Authors say "in three maternity units in the UK". This could be confusing, since it can be interpreted as representing all UK maternities (111 according to NHS Maternity Statistics, England 2016-17). It is suggested to change the term "representative" to "different" and to specify that the activity performed in these maternities may reflect the activity set of public maternities with a similar volume of annual deliveries.	We have replaced “representative” with “different”.
It is also suggested to authors to include the dates planned for the study. Start date and expected end date.	We have included start and expected end dates in the Methods section: “The study is expected to run between April 2019 and June 2020.”

Reviewer's comments	Authors' response
This is a well-written, clearly described protocol for a novel, interesting study in the area of fetal monitoring in pregnancy. I have some minor comments only which I hope are helpful;	Thank you for your comments.
1. Page 3, lines 46-47, under 'Fetal Monitoring in Labour', it might be of benefit to mention, in addition to the two main types of monitoring, when each is used/recommendations for use of each.	We have added: "NICE guidelines recommend offering intermittent auscultation to women at low risk of complication during labour; EFM is the recommended option in the presence of certain signs or conditions specified in the guidelines (such as fresh vaginal bleeding, hypertension or high temperature, or when oxytocin is used)."
2. I'm not very clear how the Systems Engineering data are extracted from the ethnographical study, and integrated; suggest insert a few 'plain language' lines to help better explain this (page 7, Lines 22-23 where you state "it seeks to ensure that all relevant aspects (social and technical) of a complex process or system are considered and integrated into a whole")	Thank you for raising this issue. Systems engineering data will not be extracted from ethnographic data. Data will be collected (through observations) by a systems engineer, consistent with standard practice in engineering research. The systems engineer's observations will therefore be separate from the observations conducted by the ethnographer. To make this clearer, at page 6 (in the section where we explain the ethnographic and systems engineering approaches) we have added the following text: "Systems engineers working on health services can collect and use multiple types of data, both quantitative and qualitative, including data collected through observation." We further clarify, on page 7, that: "We expect that a social scientist and a systems engineer will each spend up to seven days each in the three participating maternity units (together or at different times), combining day and night observations, and conducting observation blocks of around 8 hours per day/night."
3. Please provide a rationale for why doctors and midwives with < 1 year experience are excluded; especially as you state on page 8; "we will seek to interview participants with different professional backgrounds (midwives and obstetricians), seniority and professional experience....." Junior level staff could provide valuable data to this study?	In the light of this comment, we have reflected again on our sampling rationale. Our original reasoning for interviewing only more experienced clinicians was twofold: first, due to their longer work experience, they may be able to offer a more informed input on the factors that influence EFM with CTG, and second, very early career staff would still be involved in observations and informal conversation with researchers. This was not clearly explained in the paper. We have now changed our approach as we agree with your suggestion that, if very

	early career staff are heavily involved in EFM-related practice, then it would be inappropriate to exclude them from interviews. We have therefore amended the protocol text at page 6, and we plan to submit a minor amendment to the Health Research Authority to notify that we do not plan to exclude staff with less than 1 year's experience from interviews. The relevant passage in the eligibility criteria (page 6) section now reads as follows: "For the interviews, we will include doctors and midwives in the participating maternity units who are directly involved in using EFM with CTG. We do not plan to interview women and birth partners/relatives."
4. The study I understand takes place in 3 maternity units; but can you clarify will observations occur in all locations within these units where EFM is likely to take place, or focus on a single location such as labour ward? If it includes all locations within the unit, will your sample for interviews be drawn from across locations within the settings? Please clarify in your protocol	We have added the following text at page 7 (in the 'Observations' section) to clarify this point: "Our observations will focus on EFM with CTG in intrapartum care (rather than ante-natal care); we therefore expect the bulk of the observations to be conducted in labour wards. However, depending on how each unit organises admission procedures and early labour checks, and because of the practicalities of shadowing staff, observers may also occasionally visit the antenatal ward."
5. In varied places throughout the manuscript the full term 'electronic fetal monitoring' is used, whereas, in other places the abbreviated EFM is used (e.g. page 9, lines 17-18); please review for consistency of whichever you chose to use throughout	Thank you; we have now addressed this. We kept only the first instance of "electronic fetal monitoring" and abbreviated all others as EFM.
6. On page 6 you state; "The objective of this sampling strategy is to understand CTG processes and how they vary between units and regions." Where/how does this fit into your analysis and framework development?	Our aim is not to highlight differences between units per se, but to account for the variety of processes and practices, which the literature suggests may be associated with the size of units. The analysis is at the level of practice, not maternity units, so we will not directly compare units. We rephrased the relevant paragraph at page 6 to clarify: "The objective of this sampling strategy is to understand CTG processes and practices and their variations in practice."
7. References are required for the 'risk assessment models' in the following sentence; "The hazards and risks that characterize the process using human factors concepts drawn from existing risk assessment models to guide observations"	Rephrased as follows (page 7): "the hazards and risks that characterize the process using human factors concepts drawn from existing frameworks, e.g. methods and models to guide observations (e.g. the SEIPS

	framework, ³⁷ the Yorkshire framework of factors contributing to incidents in hospitals ³⁸ process modelling ³⁹ and Task analysis ⁴⁰ .”
8. Where you state; the ethnographers and the systems engineers will analyse the whole dataset separately. Can you clarify is this referring to data that are collected “together and at different times”; i.e. independently gathered data as well as collectively gathered data?	Thank you for raising this point. To clarify, we have rephrased as follow: “Data analysis will run alongside ongoing fieldwork and will be conducted in two phases, comprising initial, disciplinary-specific analyses of data, followed by an integrative analysis. In the first phase of analysis, the ethnographers and the systems engineer will analyse their observation data separately (i.e. the systems engineer will only analyse data they will have collected, and the ethnographers will only analyse data collected by ethnographers). This is because observations are expected to be dependent on the perspective and sensitising concepts used by the different observers. In this phase, ethnographers and systems engineers will additionally analyse the whole interview dataset separately. In their respective first phase analyses of observation and interview data, the researchers will adopt different but complementary approaches. [...] In the second phase of analysis, the researchers will integrate their analyses. ”

VERSION 2 – REVIEW

REVIEWER	Valerie Smith Trinity College Dublin
REVIEW RETURNED	24-May-2019

GENERAL COMMENTS	This is a re-review of a previously reviewed manuscript. The authors have attended to all of my suggested amendments and comments.
--